# Interleukin-18 and interferon-γ single nucleotide polymorphisms in Egyptian patients with tuberculosis

Noha A. Hassuna[1]*, Mohamed El Feky[2], Aliae A. R. Mohamed Hussein[3], Manal A. Mahmoud[3], Naglaa K. Idriss[4], Sayed F. Abdelwahab[1,5], Maggie A. Ibrahim[2]

**1** Department of Medical Microbiology and Immunology, Faculty of Medicine, Minia University, Minia, Egypt,
**2** Department of Medical Microbiology and Immunology, Faculty of Medicine, Assiut University, Assiut, Egypt,
**3** Department of Chest Diseases and Tuberculosis, Faculty of Medicine, Assiut University, Assiut, Egypt,
**4** Department of Medical Biochemistry, Faculty of Medicine, Assiut University, Assiut, Egypt, **5** Division of Pharmaceutical Microbiology, Department of Pharmaceutics and Industrial Pharmacy, Taif College of Pharmacy, Taif University, Taif, Saudi Arabia

* nohaanwar@mu.edu.eg

## Abstract

### Background

Interleukin-18 (IL-18) and interferon-γ (IFN-γ) are cytokines of crucial role in inflammation and immune reactions. There is a growing evidence supporting important roles for IL-18 and IFN γ in tuberculosis (TB) infection and anti-tuberculosis immunity.

### Objective

To evaluate the role of polymorphisms in IL-18-607 and -137 and INF-γ +874 in susceptibility to TB infection among Egyptian patients.

### Methods

A case control study was conducted to investigate the polymorphism at IL-18-607, -137 and INF-γ+874 by sequence specific primer-polymerase chain reaction (SSP- PCR) in 105 patients with pulmonary and extra pulmonary tuberculosis and 106 controls.

### Results

A significant protective effect against TB was found in homozygous CC genotype at IL-18 -137G/C, in addition to a 7-fold risk with GG and GC genotypes in the recessive model. Apart from a decreased risk with the AC genotype, no association was detected between the susceptibility to TB and different genotypes or alleles at the IL-18 -607A/C site. The homozygous AA genotype in INF-γ+874 showed a significant higher risk to TB than the homozygous TT or heterozygous AT genotypes with nearly a 2-fold risk of TB infection with the A allele. Regarding haplotype association, the GC haplotype was strongly associated with TB infection compared to other haplotypes.

**Data Availability Statement:** All relevant data are within the paper.

**Funding:** The authors received no specific funding for this work.

**Competing interests:** The authors have declared that no competing interests exist.

## Conclusion

These findings suggest; for the first time in Egypt; a significant risk to TB infection with SNP at the IL-18-137G/C with no LD with SNP at the IL-18-607 site. The homozygous AA genotype in INF-γ+874 showed a significant higher risk to TB than the homozygous TT or heterozygous AT genotypes.

## Introduction

Tuberculosis (TB) is a notoriously infectious disease caused by *Mycobacterium tuberculosis* (MTB) [1]. A world-wide report estimated over 8.5 million TB cases distributed all over the world with the majority of cases in Africa and Asia [2]. Progression of TB occurs due to numerous reasons such as environmental factors, malnutrition, infection by human immune deficiency virus (HIV), immunosuppressive therapy and diabetes mellitus [3]. Despite the fact that over third of the world's population is affected with TB, progression to active disease occurs only in 5–10% of these cases [3], which indicates the potentially major role of host genetic factors in the susceptibility to progression of TB. Several studies analyzed the relationship between various genetic variations and susceptibility to TB, particularly those in cytokine genes [4–6]. Cytokines play a significant role in host susceptibility and progression of TB. Interleukin -18 (IL-18), one of the IL-1 family members, is a pro-inflammatory cytokine with plays a vital role in the inflammatory cascade [7].

Production of IL-18 in response to mycobacterial antigens (Ags) is highly associated with IFN- γ production and with mycobacterial defensive immunity [8]. However, IL-18 levels were elevated in chronic refractory TB (CRTB) patients' peripheral blood mononuclear cells as well as at the Tuberculous pleurisy (TBP) site, indicating a potential role for IL-18 in both protective immunity and human TB pathological responses [9]. The IL-18 gene is located on the long arm of chromosome 11 (11q22.2-q22.3) and incorporates many genetic polymorphisms, particularly in the promotor region.

The changes in IL-18 gene promotor have an influence on IL-18 development and operation. Two of the most common single nucleotide polymorphisms in IL-18 gene are located in the promoter region: -607C/A (rs1946518) and -137G/. C (rs187238) [10, 11]. Cloning and transcriptional research revealed that IL-18 expression was modified with these two polymorphisms. Nevertheless, studies done to investigate possible contribution of IL-18 polymorphisms to TB susceptibility have produced conflicting findings [11].

Interferon gamma (IFN- γ) is a key T-helper (Th) type -1 cytokine primarily produced by natural killer cells and T cells. It plays a crucial role in the activation of the macrophage to combat MTB infection [12].

A SNP (+874T/A; rs2430561) is found at the 50- end of a CA repeat at the first human IFN-γ intron.

The +874 T allele is linked to the 12 CA repeats, while the A allele is linked to the non-12 CA repeats. This leads to a change in IFN-γ expression as T allele is associated with a higher IFN-γ expression, while A allele is correlated with low expression [13].

Multiple studies investigated the association between IL-18 and INF γ polymorphisms and susceptibility to TB. However, consistent conclusions have not been reached, and single research is unable to determine combined effects. Furthermore, to the best of our knowledge there are no previous reports regarding IL-18 promotor SNP effect on the susceptibility to TB among Egyptian patients. Therefore, the aim of this study was to determine the association of

(IL -18) and (INF-γ) polymorphisms ((-607C/A) and (-137G/C)) and (+874T/A); respectively and the risk of TB.

## Patients and methods

### Patients

This case-control study included 105 new tuberculous patients and 106 controls. Cases were enrolled from the tuberculosis outpatient clinic of Chest Department, Assiut University Hospital between December 2017, and May 2018. The study protocol was approved by the Local Ethics Committee (IRB no: 17300444) and was carried out in accordance with the Declaration of Helsinki. Written informed consents were obtained from all patients or those responsible for them.

Patients attending the outpatient tuberculosis clinic with newly diagnosed pulmonary and extrapulmonary TB were included. Pulmonary TB was confirmed by positive sputum smear and culture for MTB. Extra pulmonary TB was diagnosed by the presence of characteristic tuberculous granulomas composed of multinucleated giant cells, epithelioid cells and foamy cells surrounded by a rim of lymphocytes on histopathologic examination of biopsies from affected sites. Pott's disease was diagnosed when magnetic resonance imaging (MRI) of the spines showed irregularity of both the endplate and anterior aspect of the vertebral bodies, with bone marrow edema and enhancement; T1: hypointense marrow in adjacent vertebrae, T2: hyperintense marrow, disc, soft tissue infection, T1 C+ (Gd): marrow, subligamentous, discal, dural enhancement [14].

Tuberculous Choroiditis was characterized by deep, multiple, discrete, yellowish lesions ranging from 0.5 to 3.0 mm in diameter [15].

HIV-positive subjects, previously treated TB cases, patients who did not regularly attend the scheduled clinic visits during the six months treatment period and patients who refused to participate were excluded from this study. Our sample is considered representative of a larger population.

Controls (n = 106) were age and sex matched participants from the same region and ethnic background with no history of previous or current TB infection.

All the patients received the following treatment:

First 2 months: Rifampicin 10 mg/kg, isoniazid 5 mg/kg, pyrazinamide (according to body weight range: 18.2–25 mg/kg for 40–55 kg body weight, 20–26.8 mg/kg for 56–75 kg and 22.2–26.3 mg/kg for 76–90 kg) and ethambutol (14.5–20 mg/kg for 40-55kg, 16–21.4 mg/kg for 56–75 kg and 17.8–21.1 mg/kg for 76-90kg). Then rifampicin and isoniazid are continued for 4 months.

### Methods

**Sampling and DNA extraction.**   Venous blood samples were collected from all study participants in EDTA-coated tubes (3–4 ml) and stored at -80˚C for further use. Extraction of genomic DNA was done by standard chemical methods [16]. DNA efficiency and quantity were assessed via 0.8% agarose gel electrophoresis and by UV spectrophotometer. The extracted DNA was stored at -20˚C for further SNP analysis.

**SNP detection.**   *Detection of IL-18 genetic polymorphisms*. Sequence-specific primers PCR (SSP-PCR) method was used to detect the IL-18 -607C/A and IL-18-137G/C SNPs [17]. Briefly, a common reverse primer: (R): 5′-TAA CCT CAT TCA GGA CTT CC-3′, and two sequence-specific forward primers (F1): 5′-GTT GCA GAA AGTGTA AAA ATT ATT AC-3′ and (F2): 5′-GTT GCA GAA AGTGTA AAA ATT ATT AA-3′ were used for the IL-18

-607C/A SNP with a 196-bp fragment size. An internal positive control was added using a control forward primer: 5′-CTT TGC TAT CAT TCC AGG AA-3′ with a 301 bp fragment size.

Detection of the IL-18 -137G/C SNP was done using a common reverse primer (R): 5′-AGGAGG GCA AAATGC ACT GG-3′ and two sequence-specific forward primers (F1): 5′-CCC CAA CTT TTA CGG AAG AAAAG-3′ and (F2): 5′-CCC CAA CTT TTA CGG AAG AAA AC-3′ with a 261 bp fragment size. An internal positive control was added using control forward primer: 5′-CCA ATA GGA CTG ATT ATT CCG CA-3′ with a 446 bp fragment size.

For each position, three PCR reactions were carried out. Each reaction was performed in a total volume of 25 μL consisting of 12.5 μL MyTaq HS Red Master Mix (Bioline, USA Inc.),1 μL of each primer (the common reverse and either of the sequence-specific primers or the control primer), 30 ng genomic DNA, and nuclease-free water.

PCR conditions for IL-18-137G/C position were as follows: initial denaturation (4 min at 94˚C), followed by 5 cycles (20 s at 94˚C, 1 min at 68˚C), and 25 cycles (20 s at 94˚C, 40 s at 62˚C and 40 s at 72˚C).

PCR conditions for -607 position were as follows: initial denaturation (4 min at 94˚C), followed by 7 cycles (20 s at 94˚C, 40 s at 64˚C, and 40 s at 72˚C) and 35 cycles (20 s at 94˚C, 40 s at 57˚C and 40 s at 72˚C). The product of each reaction mixture was evaluated by electrophoresis on 1% agarose gels.

*Detection of IFN-γ genetic polymorphisms*. Amplification refractory mutation system (ARMS) reaction was used to detect IFN-γ (+874T/A) SNP [18]. The following primers were used: A allele primer: 5-TTC TTA CAA CAC AAA ATC AAA TCA-3, T-allele primer:5-TTC TTA CAA CAC AAA ATC AAA TCT-3, Generic primer: 5- TCA ACA AAG CTG ATA CTC CA -3 with a 261 fragment. Successful PCR was checked by using human growth hormone internal control primers: forward primer: 5-GCC TTC CCA ACC ATT CCC TTA-3, reverse primer: 5-TCA CGG ATT TCT GTT GTG TTT C-3 with a 429 bp fragment.

For each sample two reactions were carried in 25 μL total volume with Master Mix, DNA as mentioned above except that each reaction contained specific primer mix (1 μL of one of the two allele-specific primers and 1 μL of the generic primer) and the internal control primers mix.

The PCR conditions were carried out as follows: 1 min at 95˚C followed by 10 cycles (15 s at 95˚C, 50 s at 65˚C and 40 s at 72˚C), then 20 cycles (20 s at 95˚C, 50 s at 56˚C and 50 s at 72˚C). The PCR products were analyzed as mentioned above.

**Statistical analysis.** Data was collected and analyzed using SPSS (Statistical Package for the Social Science, version 20, IBM, and Armonk, New York). Continuous normally distributed variables were presented as means ± standard deviations, Student t-test was used to compare means. Test of significances: Chi square and Fisher Exact tests were used to compare the difference in distribution of frequencies among different groups. Multivariate logistic regression analysis was measured to examine the predictive power of the studied Cohort's genotypes & Alleles (Odds Ratio -OR-, 95% confidence interval -95% CI- and Likelihood Ratio Test–LRT-Using the expectation maximization (EM) algorithm, the Haplo View program (version 4.2) was applied to estimate the haplotypes and the linkage disequilibrium (LD). P value <0.05 was considered statistically significant.

## Results

This study included 105 TB patients: 64 females (60.9%) and 41 males (39.1%) with a mean age of 40.11 ± 16.15 years and 106 controls: 48 females (45.3%) and 58 (54.7%) males with a mean

**Table 1. Demographic and clinical characteristics of the study participants.**

| Variable | Cases (n = 105) | Controls (n = 106) | P-value |
|---|---|---|---|
| Age (Mean ± SD, years) | 40.1 ± 16.1 | 37.6± 13.6 | 0.4 |
| Sex (N, %) | | | |
| Male | 41 (39.0%) | 48 (45.3%) | 0.6 |
| Female | 64 (60.9%) | 58 (54.7%) | 0.1 |
| Smoking status | | | |
| Non-smoker (N, %) | 82 (78.1%) | 75 (70.8%) | 0.1 |
| Ex-smoker (N, %) | 21 (20.0%) | 26 (24.5%) | 0.8 |
| Current smoker (N, %) | 2 (1.9%) | 5 (4.7%) | 0.2 |
| Comorbidity | | | |
| None | 80 (76.1%) | 78 (73.6%) | 0.1 |
| Pregnancy | 1 (0.9%) | 0 (0.0%) | 0.1 |
| HTN | 13 (12.3%) | 12 (11.3%) | 1.0 |
| Malignant mesothelioma | 1 (0.9%) | 0 (0.0%) | 0.1 |
| HCV | 2 (1.9%) | 1 (0.9%) | 0.7 |
| DM | 9 (8.5%) | 12 (1.3%) | 0.4 |
| Asthma | 1 (0.9%) | 0 (0.0%) | 0.1 |
| Ischemic heart disease | 1 (0.9%) | 3 (2.9%) | 0.6 |

DM: Diabetes mellitus, HTN: Hypertension, HCV: Viral hepatitis C.

**Table 2. Diagnosis of the included TB cases.**

| Diagnosis | Frequency | Percentage |
|---|---|---|
| *Pulmonary TB | 74/105 | 70.5% |
| *Extra-pulmonary TB | 31/105 | 29.5% |
| TB pleural effusion | 11/31 | 35.5% |
| TB Lymphadenitis | 5/31 | 16.1% |
| TB choroiditis | 1/31 | 3.2% |
| TB pericardial effusion | 2/31 | 6.5% |
| Pott's disease | 1/31 | 3.2% |
| TB ascites | 2/31 | 6.5% |
| TB cystitis | 1/31 | 3.2% |
| TB enteritis | 3/31 | 9.7% |
| TB mastitis | 3/31 | 9.7% |
| TB peritonitis | 1/31 | 3.2% |
| TB synovitis | 1/31 | 3.2% |

age of 37.65± 13.6. There were non-significant differences regarding demographic and clinical characteristics of the study participants as illustrated in Table 1. Pulmonary and extra pulmonary TB represented 70.5% and 29.5%, respectively, of the enrolled cases as shown in Table 2.

## Polymorphisms and susceptibility to tuberculosis

**Polymorphism at IL-18 -607C/A (rs1946518).** The observed genotype distribution of -607C/A SNP in control was not comparable to that predicted by HWE: the observed frequency of the AA genotype was 9 observed to 23.1, AC genotype was 81 observed compared to

52.7 expected and CC genotype was 16 observed compared to 30.1 expected (P = 0.0002). In addition, the observed distribution for the same SNP was not comparable in cases to that predicted by HWE: AA genotype was 13 observed to 18.8 expected, AC genotype was 63 observed compared to 51.2 expected and the CC genotype was 29 compared to 34.8 expected (P = 0.06).

The frequencies of AA, AC and CC genotypes at position -607C/A were 8.5%, 76.4% and 15.1%; respectively in controls, compared to 12.4%, 60%, 27.6%; respectively in cases with a significant decreased risk of TB with AC both as a genotype as well as in the dominant model (OR = 0.43, 95% CI 0.22–0.88, p = 0.015 and OR = 0.47, 95% CI 0.24–0.93, p = 0.026, respectively). The frequency of C-allele in controls and patients was 53.3% and 57.6%, respectively and were considered as reference group for regression analysis while the frequency of A-allele in controls and patients was 46.7% and 42.4%, respectively (Table 3), with no possible risk with either of the alleles per se.

## Polymorphism at IL-18 -137G/C (rs187238)

The observed genotype distribution of IL-18-137G/C SNP in control was not comparable to that predicted by HWE: the observed frequency of the CC genotype was 42 observed to 27, GC genotype was 23 observed compared to 53 expected and GG genotype was 41 observed compared to 26 expected (P = 0.003). On the other hand, the observed distribution for the same

**Table 3. Genotypic & allelic frequencies of the studied cohort (cases vs. controls).**

| | | | Case (n = 105) | Control (n = 106) | OR (95% CI) | $X^2$ | P value |
|---|---|---|---|---|---|---|---|
| IL-18-607 C/A (rs1946518) | Genotypes | CC | 29 (27.6%) | 16 (15.1%) | Reference | | |
| | | AC | 63 (60%) | 81 (76.4%) | 0.43 (0.22–0.88) | 5.88 | **0.015***|
| | | AA | 13 (12.4%) | 9 (8.5%) | 0.8 (0.3–2.3) | 0.18 | 0.671 |
| | Alleles | C | 121 (57.6%) | 113 (53.3%) | Reference | | |
| | | A | 89 (42.4%) | 99 (46.7%) | 0.84 (0.57–1.23) | 0.8 | 0.372 |
| | Dominant model | CC | 29 (27.6%) | 16 (15.1%) | Reference | | |
| | | AC + AA | 76 (72.4%) | 90 (84.9%) | 0.47 (0.24–0.93) | 4.93 | **0.026***|
| | Recessive model | AA | 13 (12.4%) | 9 (8.5%) | Reference | | |
| | | AC+CC | 92 (87.6%) | 97 (91.5%) | 0.66 (0.28–1.67) | 0.85 | 0.355 |
| IL-18-137 G/C (rs187238) | Genotypes | GG | 56 (53.3%) | 41 (38.7%) | Reference | | |
| | | GC | 40 (38.1%) | 23 (21.7%) | 1.27 (0.66–2.39) | 0.53 | 0.467 |
| | | CC | 9 (8.6%) | 42 (39.6%) | 0.16 (0.07–0.37) | 21.81 | **<0.001***|
| | Alleles | G | 152 (72.4%) | 105 (49.5%) | Reference | | |
| | | C | 58 (27.6%) | 107 (50.5%) | 0.37 (0.25–0.56) | 23.14 | **<0.001***|
| | Dominant model | GG | 56 (53.3%) | 41 (38.7%) | Reference | | |
| | | GC+CC | 49 (46.7%) | 65 (61.3%) | 0.55 (0.32–0.95) | 4.56 | **0.033***|
| | Recessive model | CC | 9 (8.6%) | 42 (39.6%) | Reference | | |
| | | GC+GG | 96 (91.4%) | 64 (60.4%) | 7 (3.27–15.58) | 27.75 | **<0.001***|
| IFN-γ+874 T/A (rs2430561) | Genotypes | TT | 16 (15.2%) | 8 (7.5%) | Reference | | |
| | | AT | 60 (57.1%) | 49 (46.2%) | 0.61 (0.25–1.57) | 1.09 | 0.298 |
| | | AA | 29 (27.6%) | 49 (46.2%) | 0.3 (0.11–0.74) | 6.47 | **0.011***|
| | Alleles | T | 118 (56.2%) | 147 (69.3%) | Reference | | |
| | | A | 92 (43.8%) | 65 (30.7%) | 1.76 (1.19–2.63) | 7.81 | **0.005***|
| | Dominant model | TT | 29 (27.6%) | 49 (46.2%) | Reference | | |
| | | AT+AA | 76 (72.4%) | 57 (53.8%) | 2.25 (1.58–4.01) | 7.84 | **0.005***|
| | Recessive model | AA | 16 (15.2%) | 8 (7.5%) | Reference | | |
| | | AT+TT | 89 (84.8%) | 98 (92.5%) | 0.45 (0.19–1.09) | 3.1 | 0.079 |

SNP was comparable in cases to that predicted by HWE: CC genotype was 9 observed to 8 expected, GC genotype was 40 observed compared to 41 expected and the GG genotype was 56 compared to 55 expected (P = 0.89). The frequencies of CC, GC and GG genotypes at position -137G/C were 39.6%, 21.7% and 38.7%; respectively in controls, compared to 8.6%, 38.1%, 53.3%; respectively. A strong protective effect against TB was observed with CC genotype (OR = 0.16, P<0.001) and the C allele (OR = 0.37, 95% CI 0.25–0.56), p<0.001) (Table 3). In addition, a 7-fold increased risk for TB was found in the recessive model in those having GG and GC genotypes (OR: 7(3.27–15.58), p<0.001).

### Polymorphism at IFN-γ +874T/A (rs2430561)

The observed genotype distribution of **+874T/A** SNP in control was comparable to that predicted by HWE: the observed frequency of the AA genotype was 8 observed to 9.9, AT genotype was 49 observed compared to 45 expected and TT genotype was 49 observed compared to 40.9 expected (P = 0.67). Also, the observed distribution for the same SNP was comparable in cases to that predicted by HWE: AA genotype was 16 observed to 20.1 expected, AT genotype was 60 observed compared to 51.7 expected and the TT genotype was 29 compared to 33.1 expected (P = 0.26). The frequencies of AA, AT and TT genotypes at position **+874T/A** were 7.5%, 46.2%, 46.2%; respectively in control, compared to 15.2%, 68.9% and 27.6%; respectively in cases. The homozygous AA genotype showed a significant three-fold risk of TB infection and the heterozygous AT showed a significant two-fold risk (OR = 3.38, 95% CI 1.35–8.94, P = 0.01 and OR = 2.07, 95% CI 1.15–3.65, p = 0.016, respectively). Possession of the A allele was associated with increased risk of TB infection (OR = 1.76, 95% CI 1.19–2.63, P = 0.005) (Table 3).

Four different haplotypes existed (Fig 1a), where GC was the commonest haplotype in cases compared to healthy controls and was strongly correlated with TB susceptibility ($X^2$:11.6, p<0.008). No LD was observed between the 2 SNP studied in IL-18 as D' was 10 with $r^2$ = 0.006 (Fig 1b).

### Discussion

Tuberculosis remains the world's leading cause of mortality from infectious disease among adults, with more than 10 million newly diagnosed with tuberculosis each year. There is growing evidence suggesting an important role for inflammatory cytokines in the pathogenesis of TB [19, 20]. The presence of genotypic variations in different cytokines is known to be

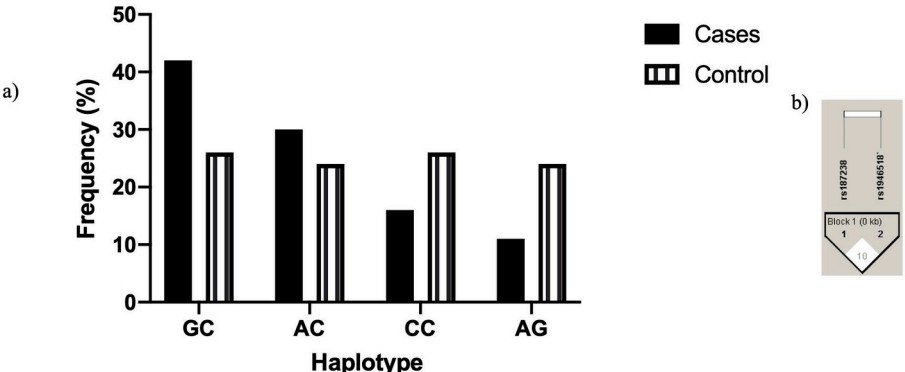

**Fig 1. Different haplotypes in rs1946518 rs187238 positions in IL-18 promotor a) Frequency of different haplotypes. b) Linkage equilibrium between the 2 SNPs in IL-18 (Haploview).**

associated with increased susceptibility to TB infection [11, 21] and hence, recognizing the genetic factors influencing TB has increasingly attracted interest. A relation may occur when the polymorphism itself is functional and leads to altered susceptibility to the disease, or when an allele SNP is in LDS with an allele which is associate with disease susceptibility, or because of a conflicting effect induced by population. Genetic markers in human leukocyte antigen (HLA) and non-HLA genes such as toll-like receptors (TLRs), cytokine/chemokines and their receptors and vitamin D receptor (VDR) have been documented in studies to predict TB susceptibility [22]. In fact, several studies have shown that genetic variations in cytokines/chemokines genes are associated with susceptibility to as TB as IFN-γ [23], IL-12 [24], IL-10 [25], and others [26].

In this study, we explored for the first time the relationship between polymorphisms in 2 cytokine genes (IL-18 and IFN- γ) and susceptibility to TB among Egyptian patients in a case–control study, which is useful for detecting disease-associated gene polymorphism stratification.

Regarding the effect of IL-18 polymorphisms on TB infection, the results obtained by previous studies were contradictory.

In this study, we found no significant risk for TB with either of the homozygous genotypes, however; the heterozygous AC genotype was associated with a slightly lower risk for TB infection. This is partially consistent with previous studies reporting the lack of significant association between SNP at the Il-18-607 position and TB susceptibility [27, 28]. Furthermore, Zhen et al. [11] conducted a meta-analysis of five case-control studies, involving 1293 TB cases and 1724 healthy subjects for the IL-18 -607C/A polymorphism and found no association between this variant and TB susceptibility.

Notably, the C allele distribution in our cohort (57.6%) is similar to those among Chinese and a Brazilian cohorts [28, 29], however it is lower than that found among South India population and Iranian population [27, 30].

A high risk for TB was detected in the current study with GG and GC genotypes for IL-18-137G/C (rs187238) SNP with a protective effect of the CC genotype.

This finding is in agreement with the evidence that IL-18-137 C allele may be in favor of increased promoter activity of IL-18 gene, with an increased level of IL- 18, which is known to be associated with more robust TB resistance and lower TB susceptibility [31]. In addition, Shen et al. [32] conducted a meta-analysis for a total of 5 studies with 558 TB patient and 720 controls. The results showed that IL-18-137G/C polymorphisms in the IL-18 gene were associated with increased TB risk in china when comparing the G allele vs. C allele (OR = 1.49, 95% CI = 1.21–1.84, P = 0.0002, GG vs. GC+CC, P = 0.0003) [32]. Interestingly, the G allele frequency in our population (72.2%) was similar to Indian (76%) [30] and Brazilian population (74%) [29], but lower than Chinese (83%) [28]. The variability of the distribution of IL-18 genes SNP between various ethnicities and population groups could explain why populations from various races show heterogenous immune responses to TB and carry different susceptibilities. It is worth noting here to mention that we found no LD between the different haplotypes of the 2 SNP in IL-18 promotor. However, a significant association was found between the GC haplotype and susceptibility to TB infection. This haplotype could be reflected in the form of lower IL-18 production and more risk of TB development.

Furthermore, we found a significant role for the IFN-γ rs2430561 SNP in the susceptibility to TB. The homozygous AA genotype was associated with increased risk for TB infection and the A allele was found a strong risk factor as well.

These results are consistent with those found among Egyptian patients in a previous study, where the AA genotype was more frequent among TB patients [33]. Furthermore, other studies found a higher risk of TB in homozygous AA patients [34, 35]. These findings are explained

by the lower IFN-γ in the AA genotype, since this SNP lies within the binding site of the NF-kappa-B transcription factor [36]. In addition, the increased frequency of AT among TB patients in our study was consistent with several studies carried on Caucasian and Asian ethnicities in USA, India, Brazil, Taiwan and China [37–41].

Since the majority of TB infections were of the pulmonary type with a scarcity of extra pulmonary TB, studying the association between the affected system and different polymorphisms studied in this work was not possible.

## Limitations

Although this study present for the first time in Egypt a possible association between IL-18-137G/C SNP and TB susceptibility, it still has some limitations like the relatively small sample size. In addition, the lack of cytokine levels measurement was not done, which was due to limited funding resources.

## Conclusion

Our results denoted a potential association between IL-18-137G/C polymorphism and human susceptibility to TB. In addition, IFN-γ+874 AA genotype was found to deliberate a defensive effect. The mechanisms of how polymorphisms of IL-18 promotor site SNP influence its protein synthesis should be further studied.

## Acknowledgments

Taif University Researchers Supporting Project number (TURSP-2020/51), Taif University, Taif, Saudi Arabia. Mohamed R Abdelrahman for revising the language of the manuscript.

## Author Contributions

**Conceptualization:** Noha A. Hassuna, Mohamed El Feky, Aliae A. R. Mohamed Hussein.

**Data curation:** Noha A. Hassuna, Manal A. Mahmoud.

**Formal analysis:** Noha A. Hassuna, Maggie A. Ibrahim.

**Methodology:** Noha A. Hassuna, Mohamed El Feky, Maggie A. Ibrahim.

**Project administration:** Mohamed El Feky, Naglaa K. Idriss.

**Resources:** Mohamed El Feky.

**Supervision:** Noha A. Hassuna, Mohamed El Feky, Aliae A. R. Mohamed Hussein, Sayed F. Abdelwahab.

**Validation:** Mohamed El Feky, Naglaa K. Idriss.

**Writing – original draft:** Noha A. Hassuna, Manal A. Mahmoud, Maggie A. Ibrahim.

**Writing – review & editing:** Noha A. Hassuna, Mohamed El Feky, Sayed F. Abdelwahab.

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
