## [Decision Letter · Decision Letter 0]

26 Oct 2020

PONE-D-20-27317

Interleukin-18 and Interferon-γ Single Nucleotide Polymorphisms in Egyptian patients with Tuberculosis.

PLOS ONE

Dear Dr. Hassuna,

Thank you for submitting your manuscript to PLOS ONE. After careful consideration, we feel that it has merit but does not fully meet PLOS ONE’s publication criteria as it currently stands. Therefore, we invite you to submit a revised version of the manuscript that addresses the points raised during the review process.

Specifically, the Authors should revise the manuscript with particular focus on the methods section, as requested by Reviewer2.

We look forward to receiving your revised manuscript.

Kind regards,

Cinzia Ciccacci

Academic Editor

PLOS ONE

Journal Requirements:

2. In your Methods section, please provide additional information about the participant recruitment method and the demographic details of your participants.

Please ensure you have provided sufficient details to replicate the analyses such as:

a) a description of any inclusion/exclusion criteria that were applied to participant recruitment,

b) a statement as to whether your sample can be considered representative of a larger population, and

c) a description of how participants were recruited.

3. Thank you for stating in the text of your manuscript "Written informed consents were obtained from all patients or those responsible for them."

Please also add this information to your ethics statement in the online submission form.

Reviewers' comments:

Reviewer's Responses to Questions

**Comments to the Author**

1. Is the manuscript technically sound, and do the data support the conclusions?

Reviewer #1: Yes

Reviewer #2: Partly

2. Has the statistical analysis been performed appropriately and rigorously? 

Reviewer #1: Yes

Reviewer #2: No

3. Have the authors made all data underlying the findings in their manuscript fully available?

Reviewer #1: Yes

Reviewer #2: Yes

4. Is the manuscript presented in an intelligible fashion and written in standard English?

Reviewer #1: No

Reviewer #2: Yes

5. Review Comments to the Author

Reviewer #1: The work deals with an essential subject of infectious diseases, including tuberculosis.

However, it needs to be revised again:

1. the linguistic correctness and clarity of the document should be checked

2. the text should be stylistically uniform and adapted to the requirements of the journal

2. it is necessary to harmonize the writing of genes with or without information on pairs of alleles

3. in the text, spaces separating numerical values from units are missing several times

4. the storage temperature of the samples should be verified

5. in the description of the results, double brackets appear several times, with no justification

6. in Table 1, the number of decimal places should be standardized

7. discussion should be enriched with additional publications analyzing SNP polymorphism in tuberculosis

Reviewer #2: - Studies on the role of genetics in the pathogenesis of susceptibility to TB disease is always interesting. In this study, authors objective is to find whether there in an association between SNP-607 C/A and -137 G/C in IL-18 gene and also SNP +874 A/T IFN-gamma gene with susceptibility to TB both pulmonary and extrapulmonary in the Egyptian population. There is no clarity regarding this matter yet. Various studies and meta-analyzes with populations from different countries, races and ethnicities show inconclusive results, although most of the results show no association. This study is quite interesting because it is the first study conducted in Egyptian and the results show that there is an association at least between SNP -137 G / C of IL-18 gene and susceptibility to suffer from TB. I have a few questions and comments as follows:

- Abstract section

o To emphasize the importance of this study, it will be better if a background subheading added in the abstract

o In methods section the study design must be mention

o There are inconsistency in conclusion, authors was not mention association between SNP in IFN gamma and TB susceptibility, while it was appear in objective and result, meanwhile authors draw a conclusion on possible association these SNPs and the development of drug resistance which was not appear in objective section and also was not back up by data in abstract results section

- Methods section

o How authors define cases? How do they diagnose pulmonary and extra pulmonary TB?

o How about inclusion criteria? is it only new TB cases or previously treated TB cases also enrolled in the study?

o How the authors define control? It was not written in the manuscript

o Authors also should explain on how matching process was done or was not done

- Results section

o Table 1 only showed basic characteristic of cases but not controls. The data is important to show the homogeneity between cases and controls. It might be possible that the difference in basic characteristics such as comorbidities between groups that increasing the susceptibility to TB

o The OR in table 2, is it adjusted or crude OR?

o Page 12 last paragraph. How drug resistance determined? Authors should also notify this in the method section

- There were several typos on write SNP, please make appropriate correction

6. PLOS authors have the option to publish the peer review history of their article (what does this mean?). If published, this will include your full peer review and any attached files.

Reviewer #1: No

Reviewer #2: No

---

## [Author Response · Author response to Decision Letter 0]

1 Dec 2020

Dear Professors,

We would like to thank you for your valuable comments that greatly improved the overall quality of our manuscript entitled “Interleukin-18 & Interferon-γ Single Nucleotide Polymorphisms in Egyptian patients with Tuberculosis.”. We are enclosing the revised manuscript with the changes tracked and highlighted (yellow) in the text. Also, comments raised by the reviewers (in italics) are outlined below, followed by our detailed responses (in regular blue font). 

Reviewer #1: 

Comment 1: The linguistic correctness and clarity of the document should be checked

Response to comment 1: We would like to thank you for this comment. The linguistic correctness and clarity were revised.

Comment 2: The text should be stylistically uniform and adapted to the requirements of the journal

Response to comment 2:

Thank you for this comment. The manuscript is now stylistically uniform as per the journal’s requirements.

Comment 3: It is necessary to harmonize the writing of genes with or without information on pairs of alleles.

Response to comment 3: As recommended by the reviewer, the writing of alleles is now harmonized.

Comment 4: In the text, spaces separating numerical values from units are missing several times.

Response to comment 4: Thank you for pointing this out. Corrections are now done.

Comment 5: The storage temperature of the samples should be verified

Response to comment 5: We appreciate this comment. The temperature is now corrected.

Comment 6: In the description of the results, double brackets appear several times, with no justification.

Response to comment 6: We thank you for this comment. Corrections are now done.

Comment 7: In Table 1, the number of decimal places should be standardized

Response to comment 7: Thank you for this comment. Decimals are fixed.

Comment 8: Discussion should be enriched with additional publications analyzing SNP polymorphism in tuberculosis.

Response to comment 8: We appreciate this comment that enriched the discussion. Additional publications regarding other SNPs are now added.

Reviewer #2:

Abstract section

Comment 1: To emphasize the importance of this study, it will be better if a background subheading added in the abstract.

Response to comment 1: We would like to thank the reviewer for this comment. A background sub-heading is now added to the abstract.

Comment 2: In methods section the study design must be mention

Response to comment 2: Thank you for this comment. Study design is added to the methods section.

Comment 3: There are inconsistency in conclusion, authors was not mention association between SNP in IFN gamma and TB susceptibility, while it was appear in objective and result, meanwhile authors draw a conclusion on possible association these SNPs and the development of drug resistance which was not appear in objective section and also was not back up by data in abstract results section.

Response to comment 3: We apologize for this inconsistency. 

Regarding the SNP in IFNg: we now pointed to its effect in the conclusion section. Regarding the association of these SNPs and the development of drug resistance: we agree with the reviewer and thus we removed this from the conclusion due to the small number of resistant cases detected in this study.

Methods section

Comment 4: How authors define cases? How do they diagnose pulmonary and extra pulmonary TB?

Response to comment 4: Thank you for bringing this crucial point up. Definition of cases and methods of diagnosis of pulmonary and extra pulmonary cases were added to Methods section.

Comment 5: How about inclusion criteria? is it only new TB cases or previously treated TB cases also enrolled in the study?

Response to comment 5: We appreciate this comment. Inclusion criteria were mentioned in the Methods section. "Only newly diagnosed TB cases were included in the study" this was added to the Methods section.

Comment 6: How the authors define control? It was not written in the manuscript.

Response to comment 6: Thank you for this comment. Definition of controls was mentioned in the Methods section.

Comment 7: Authors also should explain on how matching process was done or was not done.

Response to comment 7: Thank you for this comment. Matching was done for age and sex and this is now added in the Methods Section.

Results section

Comment 8: Table 1 only showed basic characteristic of cases but not controls. The data is important to show the homogeneity between cases and controls. It might be possible that the difference in basic characteristics such as comorbidities between groups that increasing the susceptibility to TB

Response to comment 8: We appreciate this comment that helped us to better present our data. Demographic characteristics of controls are added and compared to those of cases with estimation of the presence of statistically significant differences.

Comment 9: The OR in table 2, is it adjusted or crude OR?

Response to comment 9: Thank you for this comment. The used OR is crude OR.

Comment 10: Page 12 last paragraph. How drug resistance determined? Authors should also notify this in the method section.

Response to comment 10: Thank you for this comment. We removed data regarding resistance due to the very small (non-significant) number of resistant cases.

Comment 11: There were several typos on write SNP, please make appropriate correction.

Response to comment 11: Thank you for this comment. Typing errors were corrected.

---

## [Decision Letter · Decision Letter 1]

21 Dec 2020

Interleukin-18 and Interferon-γ Single Nucleotide Polymorphisms in Egyptian patients with Tuberculosis.

PONE-D-20-27317R1

Dear Dr. Hassuna,

We’re pleased to inform you that your manuscript has been judged scientifically suitable for publication and will be formally accepted for publication once it meets all outstanding technical requirements.

Kind regards,

Cinzia Ciccacci

Academic Editor

PLOS ONE

Additional Editor Comments (optional):

Reviewers' comments:

Reviewer's Responses to Questions

**Comments to the Author**

1. If the authors have adequately addressed your comments raised in a previous round of review and you feel that this manuscript is now acceptable for publication, you may indicate that here to bypass the “Comments to the Author” section, enter your conflict of interest statement in the “Confidential to Editor” section, and submit your "Accept" recommendation.

Reviewer #2: All comments have been addressed

2. Is the manuscript technically sound, and do the data support the conclusions?

Reviewer #2: (No Response)

3. Has the statistical analysis been performed appropriately and rigorously? 

Reviewer #2: (No Response)

4. Have the authors made all data underlying the findings in their manuscript fully available?

Reviewer #2: (No Response)

5. Is the manuscript presented in an intelligible fashion and written in standard English?

Reviewer #2: (No Response)

6. Review Comments to the Author

Reviewer #2: (No Response)

7. PLOS authors have the option to publish the peer review history of their article (what does this mean?). If published, this will include your full peer review and any attached files.

Reviewer #2: No

---

## [Editor Report · Acceptance letter]

28 Dec 2020

PONE-D-20-27317R1 

Interleukin-18 and Interferon-γ Single Nucleotide Polymorphisms in Egyptian patients with Tuberculosis 

Dear Dr. Hassuna:

I'm pleased to inform you that your manuscript has been deemed suitable for publication in PLOS ONE. Congratulations! Your manuscript is now with our production department. 

Kind regards, 

on behalf of

Dr. Cinzia Ciccacci 

Academic Editor

PLOS ONE